# Friendly Residential Environments and Subjective Well-Being in Older People with and without Help Needs

**DOI:** 10.3390/ijerph192315832

**Published:** 2022-11-28

**Authors:** María Ángeles Molina-Martínez, Sara Marsillas, María Sánchez-Román, Elena del Barrio

**Affiliations:** 1Faculty of Psychology, Universidad Nacional de Educación a Distancia (UNED), 28040 Madrid, Spain; 2ENCAGEn-CM R&D Programme, Faculty of Education and Psychology, Francisco de Vitoria University, 28223 Madrid, Spain; 3Matia Institute of Gerontology, 28020 Madrid, Spain; 4ENCAGEn-CM R&D Programme, Research Group on Ageing (GIE-CSIC), Institute of Economics, Geography and Demography (IEGD-CSIC), Spanish National Research Council (CSIC), 28037 Madrid, Spain

**Keywords:** age-friendly cities, well-being, need of help, older people

## Abstract

Previous studies have shown that friendly environments are associated with well-being and higher quality of life in older people. This study aimed to investigate the relationship between friendly environments and subjective well-being by segmenting the population according to the need for help in performing activities of daily living (ADLs) in a representative sample of people over 55 years of age in the Basque Country (Spain) (n = 2760). To determine the predictive power of friendliness on subjective well-being, two separate linear regression models were obtained according to the need for help in ADLs. The results obtained show a greater explanatory power of the model in the case of people who required help. However, in the case of people who do not need help, subjective health had a greater weight in the predictions. This paper’s findings support the greater importance of the characteristics of the physical and social environment, as people’s functional status worsens, with friendliness being an explanatory factor for people’s well-being as they age and their dependency increases.

## 1. Introduction

In demographic rate terms, the number and proportion of older people have increased worldwide, along with life expectancy [1], with no certainty that the additional years of life are being lived in good health [2]. In this regard, the increase in longevity has been observed to have occurred due to the increase in life expectancy free of severe disability, which is being delayed, although there is no increase in life free of morbidity and mild problems, situations which people continue to live with in the final stages of life [3]. As people age, the likelihood of care dependency—understood as the loss of functional capacity that implies the need for help from others to perform basic activities of daily living—increases [4]. Current research indicates that nearly half of community-dwelling people aged over 70 years are frail [5].

The origin of environmental gerontology is based on the importance of the relationship between the individual and the environment, and how this influences and is associated with well-being in old age [6,7,8,9,10]. The document “*Active ageing: a policy framework*” highlighted that “*age-friendly physical environments can make the difference between independence and dependence*”, alluding to the idea of the environment’s importance in people’s quality of life [11]. Several authors have found evidence to support this idea [12,13,14,15,16,17].

The age-friendly cities and communities movement, launched by the World Health Organization, emerged in this context and right now is probably the international meeting point for discussing innovative public policies on ageing aimed at adapting to global demographic changes and global urbanization [18,19]. Indeed, this paradigm shift is often seen as a result of several trends, including the complexity of demographic change, the policy objective to support keeping people in their homes for as long as possible [20], and the recognition of the role of the environment in active and healthy ageing [21,22]. Age-friendliness is a holistic concept [23].

An age-friendly city is defined as “*one that encourages active ageing by optimizing opportunities for health, participation and safety in order to improve the quality of life of people as they age (...) adapts its structures and services to be accessible and inclusive of people with diverse needs and abilities*” [24].

The characteristics of an age-friendly city that contribute to the well-being of older persons include eight domains: social participation, respect and social inclusion, civic participation and employment, communication and information, health and community services, outdoor spaces and buildings, transport and housing [24]. Advancing these policies that improve people’s quality of life and well-being and enable them to remain in their homes for as long as possible necessarily involves encouraging the development of age-friendly cities [25,26] that foster these features. In 2010, the WHO launched the Global Network of Age-friendly Cities and Communities [27]. Since its inception, this network has added more than 1333 cities and communities in 47 countries [28].

Despite initiatives to improve the suitability of cities for older people, there is very little empirical research assessing how cities’ characteristics can influence the quality of life of this older population [29]. Available international evidence shows that age-friendly environments are associated with higher quality of life [30,31], higher life satisfaction [32], and greater well-being [33,34]. However, much environmental gerontological research has focused on indoor environments, mainly homes, and has ignored neighborhoods and communities [32,35,36]. Some research has been concerned with analyzing differences in friendliness in relation to gender [35], age, type of cohabitation [37], fragility [36], or rural environment [38]. Nevertheless, a central aspect of friendliness is accessibility and inclusion of people with diverse needs and abilities. When dependency sets in, relationships with the environment become even more decisive in predicting the course of ageing [39].

Spain is one of the European countries with the highest life expectancy, where older people represent 20% of the population, and will keep on rising to 36.8% in 2050 [40]. In the coming years, we will face another challenge in the form of the dramatic increase in the 80–90 age group. Data from the OECD estimate that in Spain, the population over 85 years of age will rise from 1,571,508 in 2020 to 3,309,443 in 2050 [41]. Spain is divided into autonomous regions; one of them is the Basque Country, where 22% of the population are people over 65 years old [42], while 61% of people over 55 years old live in urban areas. Moreover, 27.1% of the population over 55 years of age need help with ADLs [43]. The study of older people’s need for help dates back to the beginning of the 20th century, to the research on the capabilities, autonomy, and independence of people carried out in the fields of Medicine and Psychology. However, it was not until after the 1950s that a certain consensus was established on the meaning of the “needs of help” of older people, and the extent of their daily life. Katz, et al. [44] established a classification of basic ADLs that defines the dependency of older adults in terms of a series of self-care activities, including bathing, feeding, dressing, transferring (moving independently from bed or chair), walking, and using the toilet. This classification has been applied extensively in research, although other dimensions have been added over the years [45]. For example, Lawton and Brody [46] pointed out the importance of including other relevant areas to measure the person’s independence that go beyond the physical dimension. These areas have to do with mobility and instrumental activities of daily living (IADLs) necessary for living at home: the ability to use the telephone, shopping, cooking, laundering, housekeeping, using transportation, taking medicines, and handling finances.

The level of dependence in the performance of all these basic and instrumental activities of daily living has an influence on the relationship of older people with their physical and social environment. Therefore, it seems important to identify whether the equipment and resources available in their environment can be a relevant factor in meeting the needs for assistance that the person develops over the years.

Since 2009, the Basque Country has been part of the Age-Friendly Cities and Communities initiative when the first city in this region (Donostia-San Sebastian) joined. Currently, 70 municipalities are working towards age-friendliness in the Basque Country under the umbrella of the Euskadi Lagunkoia (Age-Friendly Basque Country) project.

The purpose of this study was to investigate the relationship between the friendliness of the residential environment and mental well-being in a representative sample of the older population in the Basque Country (Spain), segmenting the results according to people’s need for help. The proposed working hypothesis is that the features of the physical and social environment become more important as people’s functional state worsens and they need help with ADLs, with the characteristic variables of friendliness being explanatory factors of people’s subjective well-being as they grow older and their dependence increases.

## 2. Methods

### 2.1. Sample

The methodology of this study was based on a survey of a representative sample of community-dwelling residents aged 55 and over in the Basque Country. Structured interviews were conducted using a computer-assisted, questionnaire-based telephone survey. Sample selection involved using stratified random sampling, considering geographic area, age group (55–64 years old, 65–79, and 80 and over), and gender as main stratification criteria. Sample distribution followed a proportional method for territory strata and the three capital cities, and quotas were applied according to age group (55–64, 65–79, and 80 and over) and gender. Households in each stratum were chosen by randomly selecting those with one person aged 55 and over, only interviewing one person per household. Sample size was determined by the required level of disaggregation, with a minimum of 400 interviews per capital. The survey was conducted between July and September 2020, months in which the lockdown, motivated by the COVID-19 crisis, had ended in Spain, so during the phase of “new normality” [47].

The sample who responded to the complete questionnaire included in this study consisted of 2760 people (1233 men and 1527 women). All participants were informed of the aims of the study and their rights, how the data would be used and its purposes, as well as the anonymity and management of the data. All of them gave their informed consent for inclusion before they participated in the study. Ethical review and approval were waived for this study, due to the data collected in the study were anonymous and according to the Organic Law on Personal Data Protection and guarantee of digital rights (article 2.2. LOPD 3/2018). The details about how consent was collected and a sample consent form provided before the telephone survey can be found in the appendix of the article (Appendix A).

### 2.2. Variables

The variables included in this study are need of help, mental well-being, friendliness variables, social participation, stereotypes balance, civic participation and employment, communication and information, and sociodemographic variables.

Need of help—Need of help was defined in this study as a dichotomous variable based on the need of help for ADLs and IADLs of participants.

Subjective Well-being—Subjective well-being was measured using the Spanish version of the World Health Organization Five Well-Being Index (WHO-5) [48]. This 5-item index asks how the person has been feeling in the previous weeks (e.g., I have been feeling cheerful and in good spirits), with six response options and scores: all of the time (5), most of the time (4), more than half of the time (3), less than half of the time (2), some of the time (1), and at no time (0). The total raw score ranges from 0 to 25 and the final scores are obtained by multiplying the raw score by 4, with 0 representing the worst well-being and 100 representing the best imaginable well-being.

Friendliness variables—A range of 15 items was included to reflect the level of friendliness of the municipality and closer environment, according to the eight domains which define a friendly environment. Open spaces and buildings were measured by collecting data on access to four services (bank; cinema, theatre or cultural center; park or green area; and supermarket). Additionally, health care center access was assessed to measure community and health services. These items have four response options ranging from “very difficult” to “very easy”. The higher scores mean easier access.

The housing dimension was assessed by measuring the presence of obstacles or physical barriers when moving inside their home, accessing their building, and moving around the immediate environment. Furthermore, a direct question regarding obstacles to using public transport was included to assess transport. In both cases, the response options were “yes” and “no”.

The social participation domain was measured by including outdoors activities such as cultural activities, social activities, physical activities, religious acts, and tourism. A dichotomous variable was created based on whether people engaged in outdoor activities or not, using the same procedure as for indoor activities. Respect and social inclusion was measured, based on the stereotype balance and the neighborhood feeling. This concept was measured with the sense of neighborhood subscale [49]. It consists of seven items (e.g., I would be really sorry if I had to move away from the people in my neighborhood) with five response options, from strongly disagree to strongly agree. Responses were scored from 1 (strongly disagree) to 5 (strongly agree), and the total scores range from 5 to 35.

Stereotype balance was calculated based on the difference between positive stereotypes towards older adults and negative ones. Participants responded to 12 pairs of items depending on whether they consider older adults independent or dependent, productive or unproductive, healthy or sick, tolerant or intolerant, resistant or fragile, progressive or conservative, active or passive citizens, protected or helpless, sexually active or passive, integrated or marginal, non-conflictive or conflictive, and sociable or reticent. Quantitative variables were created for positive and negative stereotypes, and afterwards, a difference was calculated to obtain the stereotype balance.

Civic participation and employment was evaluated through two different variables. First, one variable asked about volunteer activities such as participation in social and community services, educational, cultural or sports associations, or social movements. A second variable asked about political participation, such as involvement in political party or trade union meetings, participation in a demonstration, or signing a petition.

Finally, communication and information was assessed through a variable asking about internet access, including mobiles or tablets.

Participants’ information was collected by including a group of demographic characteristics, namely gender, age, educational level, living environment, and marital status. Other variables of interest were perceived health status (ranging from very poor to very high) and limitations in daily activities due to health problems in the last 6 months.

### 2.3. Procedure

In order to test our hypotheses, multivariate linear regressions were selected to test which variables predict subjective well-being. After checking assumptions, two different models were tested, with the first one focusing on the complete sample and the second one selecting those older adults who need help. Each linear regression consisted of two models. Independent variables were introduced in the first model, while control variables (namely gender, age, marital status, educational level, and habitat) were introduced in the second one.

## 3. Results

The descriptive results of the main variables contained in this study are shown in Table 1. 

Overall, 27.1% of the sample needed help with basic activities of daily living.

The mean score on the well-being scale was 68.5 (SD = 20.2), indicating medium–high values for this dimension. The greatest reported access difficulties were at the cinema, theatre or cultural center, followed by the bank and the health center. Accessing a building was identified as the main physical obstacle or barrier, according to frequency (13.2%).

The mean score obtained in the subscale of sense of neighborhood was 26.71 (SD = 4.7), indicating high values for this variable.

Table 2 shows both the predictors of well-being in people who do not need help with daily activities and in people who do. Eleven significant variables regarding people who do not need help were entered into the model (R2 = 0.15; F = 38.13; df = 11), whereas nine variables were statistically significant and were entered into the model for people who need help. In this group, the model explains a higher percentage of the variance (R2 = 0.35; F = 18.83; df = 9). Five variables were entered into both models, namely subjective health, sense of neighborhood, stereotype balance, access to parks and green areas, and outdoor activities. Nevertheless, their relevance in the prediction of subjective well-being varied. In the case of people who do not need help, the most relevant variable was subjective health (estimate = −0.27, *p* < 0.05), whereas in the model of people who do need help, the most relevant one was the sense of neighborhood (estimate = 0.30, *p* < 0.05), followed by outdoor activities (estimate = 0.25, *p* < 0.05), subjective health (estimate = −0.21, *p* < 0.05), and access to parks and green areas (estimate = 0.21, *p* < 0.05). Regarding the other variables, in this model, access to the supermarket (estimate = −0.11; *p* < 0.05), the absence of obstacles when moving inside the home and when accessing the building (both estimate = 0.11; *p* < 0.05), and volunteering (estimate = 0.10; *p* < 0.05) were related to a higher subjective well-being. Here, no socio-demographic variables were significant. Regarding the model of people who do not need help, the other included variables were access to the bank (estimate = 0.04; *p* < 0.05) and to the internet (estimate = −0.04; *p* < 0.05) and perceived limitations in daily activity (estimate = 0.05; *p* < 0.05), which were related to subjective well-being. Adding gender and marital status improved the model slightly (R2 = 0.14 and R2 = 0.15, respectively).

## 4. Discussion

This paper aimed to investigate the relationship between friendliness and subjective well-being in a representative sample of older adults in the Basque Country, segmenting the results according to the need for help of the people who participated in the study.

Well-being can be considered a measure of social progress, associated with better health outcomes, better physical and cognitive function, lower levels of frailty and disability, and lower mortality [50,51,52,53,54]. The physical and social characteristics of the environment can be protective or detrimental to older people’s health [22,55]. Ultimately, friendly environments can contribute to improving older people’s well-being by matching environmental resources to individual needs.

Based on the recognition that community-dwelling older people have varying preferences, needs, and resources, the WHO advocated that a given city should accommodate this heterogeneity by “adapting its structures and services to be accessible to and inclusive of older people with varying needs and capacities” [24].

Research has shown that age-friendly environments are associated with higher levels of well-being and quality of life in older people [30,33,56]. It has even been concluded that older people who perceive their environments as age friendly are almost four times more likely to report a better quality of life than those who report lower levels of age friendliness [30]. Its positive association with well-being is not surprising, as the criteria for an age-friendly environment align almost perfectly with both concepts [33]. Park and Lee [32] found that after controlling for demographic covariates, physical and social environment features are significantly related to the life satisfaction of older people in Korea.

As people age, the likelihood of increased frailty increases, which in turn increases their needs for neighborhood features that enable them to age in place [57,58]. Thus, segmented analysis according to the need for help required can provide clues regarding the importance of different variables depending on the person’s status [59].

The models presented showed different levels of explained variance depending on the person’s need for help. Thus, when help is not needed, the explanatory power of the model is lower, which may point to a greater importance of the environment’s physical and social characteristics when people’s functional status worsens [60,61]. In the first model (without the need for help), variables related to the characteristics of a friendly environment weighed less in the predictors than personal characteristics—subjective health.

In the model obtained for people requiring help in performing ADLs, the proportion of variance explained was greater. The variable that weighed the most in the predictors was the sense of neighborliness [49]. Literature congruent with this finding has identified links of neighborhood to physical health [62], mental health [63], and well-being [64]. A study by Cramm and Nieboer [65] concluded that social cohesion and belongingness predict the well-being of community-dwelling older people in the Netherlands. Recent research places increasing emphasis on creating age-friendly cities and neighborhoods that foster a sense of place [66]. Although neighborhood conditions and individual functional ability are important [67], subjective feelings about a neighborhood can be a significant source of satisfaction, independent of objective measures of suitability or safety [68,69]. Given their declining social networks and reduced mobility [70], the neighborhood context gains importance in meeting older people’s daily needs [36,68]. The neighborhood can become an important source of social identity [71], where good social ties with neighbors are valued [36,72,73] as they contribute to satisfaction with the neighborhood [74].

Outdoor activities (cultural activities, physical activities, social activities, attendance to religious acts and tourism), although included in both models, also contribute most to explaining the level of well-being in people who need help. These types of activities become particularly relevant for older people when activity inherent to employment or responsibilities deriving from maternity/paternity disappear or are reduced and there is much more free time as a result. Therefore, the activities they spend their time on provide new references and meanings [75]. The link between participation in social and leisure activities and well-being has also been evidenced in several studies (e.g., [76]). Early studies that found this relationship were the basis for the development of activity theory in old age [77], which despite its limitations, continues to influence research and theory today [78]. Adams, Leibrandt and Moon [78] review of leisure and social activities highlighted that most studies showed positive associations between activity participation and psychosocial well-being, health, or survival. Other authors highlight how social participation mitigates loneliness and benefits older people’s health and quality of life, suggesting that social participation enhances people’s ability to age in place [36].

Easy access to parks and green spaces was also a significant variable for the subjective well-being of people who need help. Open-air spaces represent the sphere of sociability, the public domain, and the connection between the individual and the group. The configuration of these spaces is essential for the promotion of citizens’ participation in the daily life of the municipality and, ultimately, active aging. “The outdoor environment and public buildings have a major impact on the mobility, independence and quality of life of older people” [24]. Much research on the physical environment has examined physical activity levels and health issues among older people [79], identifying important attributes such as sufficient green space, accessible buildings, and adapted streets and crossings [36]. In addition, research shows that access to natural environments is described as also being essential for general well-being in people with dementia [80].

Participation in voluntary activities is also a predictor of subjective well-being for people in need of help. Several studies have found that older people’s participation in socially productive activities is associated with well-being [81,82], as well as meeting service needs in the community [36]. For example, Van Willigen [83] reported that older volunteers (aged 60+) experienced greater increases in life satisfaction and improvements in perceived health than younger people. Additionally, Freedman, et al. [84] found a positive relationship between volunteering activities and subjective well-being among people with disabilities.

In segmented models, in access to services (leisure, health center and supermarkets), people who need help with performing ADLs were more affected by ease of access compared to those who do not need help, affecting their subjective well-being. These data support the thesis of how a friendly city should adapt its structures and services to be accessible to people with diverse needs and abilities [24]. Increased dependency may prevent access to resources needed in less favorable environments. Therefore, providing services in the near and accessible environment can help maintain autonomy and independence and contribute to people’s well-being. Easily accessible proximity services can mitigate the difficulties prompted by limited mobility [85]. In this regard, a study in the United States found a significant relationship between well-being and accessibility to neighborhood services among older people living in ten cities [86]. In the United Kingdom, it was found that accessibility was a main predictor of older people’s quality of life and that facilitating access to local facilities and transport promotes independence [87]. Older people residing in communities with good accessibility to services have been found to have better physical health, quality of life, and well-being compared to those living in less accessible environments [88]. These authors further describe how empirical research highlights the association between accessibility to services and improvements in a range of quality of life outcomes for older people, such as fewer symptoms of depression, greater life satisfaction, and higher scores on overall quality of life assessments.

Accessibility of local services is an important indicator for age-friendly communities [68]. In such environments, “ageing in place” occurs more easily and to a greater degree than in neighborhoods with a shortage of services [89]. Thus, accessibility to proximity services not only facilitates independence in obtaining necessary goods and care, but also fosters a sense of living in a hospitable environment, which, in turn, promotes feelings of social inclusion [85,88].

Differences were also found in the explanatory power of indoor barriers and building access as a function of segmentation by need for assistance. Older people with functional limitations may attach greater importance to these conditions as they can help them cope with their loss of functionality. The built environment of housing has a profound impact on well-being on a variety of levels, including the social, psychological, and physiological ones [90,91]. The home is associated with individual and family biography, as well as permeating our identity [92,93]. Homes are physical environments, but they also function on a social and symbolic level in an interconnected way, as housing options also allow one to maintain ties with one’s family and friends [69]. In this sense, and also in the accessibility field, environmental gerontology since Lawton [67,94] emphasizes the role of interaction between personal competence and the physical environment of the home with the well-being of older people, showing how changes in the home (such as the removal of obstacles or the introduction of mobility aids) can improve independence. Building friendly places and inclusive neighborhoods also involves modifying existing housing, designing fully age-friendly housing, and creating accessible neighborhoods with adequate service provision [95].

## 5. Conclusions

Making cities and communities age-friendly ensures that they are inclusive and equitable places, leaving no one behind, especially the most vulnerable older people [96]. In line with previous research [36,97], this study suggests that person–environment fit is not static, given that both communities and older people change.

The limitations of our research include its cross-sectional nature, which hinders drawing causal conclusions. Longitudinal studies are needed to examine the relationship between well-being and ageing in place, which would allow us to probe the dependency relationships between environment, functionality, and well-being. This study, despite having a representative sample, was conducted in a single autonomous region, with particular characteristics, and the conditions of each region would have to be taken into account to understand differences between diverse environments and their effects on ageing people’s well-being.

Our findings have implications for the design of public policies based on a commitment to age friendliness. Influencing older people’s well-being entails investing in the environment’s social and physical conditions, especially in the case of people who need help. Public policies aimed at well-being friendliness as an outcome measure should be evaluated, because populations are ageing rapidly and this measure has been shown to correlate with other measures that place a significant burden on the community.

## Figures and Tables

**Table 1 ijerph-19-15832-t001:** Descriptive statistics.

	N	%	M	SD	
Sociodemographic Variables					
Age (55–101)	2758		69.7	10.2	
Female (vs. male)	2760	55.3			
Educational level	2728				
Below primary		14.6			
Primary		24.7			
Secondary and higher		60.7			
Living environment	2760				
Less than 20,000 inhabitants		38.9			
Between 20,000–50,000 inhabitants		12.7			
More than 50,000 inhabitants		48.5			
Married or co-living (vs. others)	2754	58.6			
Need of help, yes (vs. no)	2758	27.1			
Subjective well-being (0–100)			68.5	20.2	
Friendliness variables					
Open spaces and buildings					
Access to bank	2753				
Very difficult		3.7			
Difficult		10.1			
Easy		54.6			
Very easy		28.7			
Not used		2.9			
Access to cinema, theatre, cultural centre	2753				
Very difficult		4.4			
Difficult		15.3			
Easy		44.5			
Very easy		17.1			
Not used		18.8			
Access to park or green area	2757				
Very difficult		0.5			
Difficult		3.2			
Easy		53.1			
Very easy		41.4			
Not used		1.8			
Access to supermarket	2755				
Very difficult		1.3			
Difficult		7.0			
Easy		52.9			
Very easy		37.4			
Not used		1.4			
Health and community services					
Access to health care centre	2757				
Very difficult		1.3			
Difficult		9.4			
Easy		55.7			
Very easy		33.1			
Not used		0.4			
Housing					
Obstacles or physical barriers, yes (vs. no)					
Moving inside their home	2757	4.3			
Accessing their building	2755	13.2			
Moving around the immediate environment	2752	10.0			
Transport					
Obstacles or physical barriers, yes (vs. no)					
Using public transport	2687	7.8			
Social participation					
Indoor activities, yes (vs. no)	2760	99.7			
Outdoor activities, yes (vs. no)	2760	97.4			
Respect and social inclusion					
Stereotype balance (−12–12)	2760		−0.09	5.3	
Sense of neighbourhood (0–35)	2760		26.71	4.7	
Civic participation and employment					
Volunteer activities (0–3)	2760		0.1	0.3	
Political participation (0–2)	2760		0.1	0.4	
Communication and information					
Access to Internet, yes (vs. no)	2760	78.2			

**Table 2 ijerph-19-15832-t002:** Multivariate regression on subjective well-being (standardized regression coefficients).

	No Need of Help				Need of Help			
	Beta	Sig.	IC 95%		Beta	Sig.	IC 95%	
Limitations in daily activity	0.05 *	0.010	0.43	3.77	0.01	0.878		
Subjective wellbeing: health	−0.27 *	0.000	−8.11	−6.01	−0.21 **	0.000	−7.68	−2.69
Friendliness variables								
Respect and social inclusion								
Sense of neighborhood	0.10 *	0.000	0.24	0.56	0.30 **	0.000	0.89	1.75
Stereotype balance	0.10 *	0.000	0.23	0.49	0.10 **	0.045	0.01	1.05
Outdoor spaces and buildings								
Access to the bank	0.04 *	0.038	0.06	2.10	0.03	0.561		
Access to cinema, theatre, or cultural center	−0.04	0.092			0.00	0.956		
Access to parks and green areas	0.07 *	0.002	0.80	3.47	0.21 **	0.000	3.11	9.65
Access to the supermarket	0.03	0.317			−0.11 **	0.040	−6.19	−0.15
Health and community services								
Access to health center	0.03	0.234			0.04	0.502		
Social participation								
Indoor activities					−0.01	0.899		
Outdoor activities	0.07 *	0.000	7.19	22.80	0.25 **	0.000	11.27	24.93
Housing								
Obstacles when moving inside the home	0.02	0.324			0.11 **	0.035	0.66	18.41
Obstacles when accessing the building	0.02	0.257			0.11 **	0.028	0.70	12.26
Obstacles when moving around the immediate environment	0.007 *	0.727			0.00	0.993		
Transport								
Obstacles when moving about in public transport	−0.01	0.483			0.05	0.352		
Civic participation and employment								
Volunteering	−0.01	0.518			0.10 **	0.039	0.84	31.99
Political participation	0.002	0.900			−0.01	0.844		
Communication and information								
Access to the internet	−0.04 *	0.003	−3.92	0.17	0.00	0.937		
Sociodemographic variables								
Gender	−0.09 *	0.000	−4.81	−1.94	0.02	0.662		
Marital status	−0.05 *	0.007	−3.60	−0.56	0.00	0.974		
Age	0.02 *	0.300	0.03	0.20	0.04	0.382		
Educational level	0.03	0.228			−0.04	0.440		
Habitat	−0.028	0.154			−0.02	0.688		

* Variables included in the final model with R^2^ = 0.15. ** Variables included in the final model with R^2^ = 0.35.

## Data Availability

Restrictions apply to the availability of these data. Data was obtained from Basque Country Government and are available from the authors with the permission of Basque Country Government.

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
