# Peer review of "Friendly Residential Environments and Subjective Well-Being in Older People with and without Help Needs"

_ijerph, 2022, doi:10.3390/ijerph192315832_

Round 1
Reviewer 1 Report
Hello,
An important topic, with some revisions required: 1. line 46 (I would say age-friendly, not 'friendly cities movement'. 2. editing throughout - examples include line 151 'in'; line 26 'demographic rate'; Table I - 'park o green'. 3. there was no statement on how/whether an ethics protocol was submitted, to whom, and whether it was approved. This is why I downgraded my rating. 3. citations could include work by V. Menec.
I would like to see a clearer discussion of the variables in the final discussion, with reference to some age friendly literature from around the world, and atleast a brief comparison to this study. Otherwise, a good article.
Author Response
We are enclosing the revised manuscript of our paper "Friendly Residential Environments And Subjective Well-Being In Older People With And Without Help Needs” (ijerph-1980314). We are very grateful for the comments of the reviewers and believe that the manuscript has benefited greatly from their suggestions and feedback. Below, we respond in detail on how we have dealt with each of the comments and suggestions. The reviewer’s comments are in italic followed by our answers. In the paper the changes can be found as they are inserted with change tracker.
- line 46 (I would say age-friendly, not 'friendly cities movement'.
We have included the suggestion of the reviewer in the corresponding line.
- editing throughout - examples include line 151 'in'; line 26 'demographic rate'; Table I - 'park o green'.
We have reviewed and edited the text as suggested.
- there was no statement on how/whether an ethics protocol was submitted, to whom, and whether it was approved. This is why I downgraded my rating.
We have included a statement about the ethics protocol submission for this study.
- citations could include work by V. Menec.
We have included research conducted by V. Menec as suggested.
I would like to see a clearer discussion of the variables in the final discussion, with reference to some age friendly literature from around the world, and at least a brief comparison to this study. Otherwise, a good article.
We have also rewritten the results and discussion section, in order to provide more content and to discuss with reference to age-friendly literature as suggested.
Reviewer 2 Report
1. The purpose of the article was well developed in the introduction.
2. The sample was especially strong, being representative.
3. However, no mention is make of what ethical procedures were followed in the research.
4. As well, the research was conducted during the beginning of COVID, and I recall that Spain was badly affected by this. This should be mentioned.
5. I would have appreciated a better development of the "need for help" variable in the introduction. It is mentioned but not defined, especially not supported as a dichotomy.
6. In Tables 1 and 2, the variables mentioned do not exactly match those listed in the Method. It would help if Table 1 was sub-classified according to categories in the Method. As well, variables were broken down and used individually rather than as a whole scale total.
7. In Table 2, the source of the R squared values needs to be clarified. The asterisks have no reference point. I take it the values being referred to are for the full models including the extra variables.
8. The authors discuss differences between regression models without performing any significance testing. In most cases where a confidence interval is provided, there is no difference among slope values. There is no evidence of added significant prediction provided for the demographic variables.
In comparing the two models, the authors can safely limit their discussion to variables the "were" and "were not" significant in either model.
I suspect the overall prediction was significantly stronger in the model for those with needs. An F value can be calculated to test this using the "predicted variance" in each model and the associated df.
9. It is perhaps rather obvious that those who need help will respond systematically to access and availability variabilities in judging their wellbeing. Before I broke my leg I never considered elevators in a two-storey building important to my wellbeing. The authors get to this point eventually.
10. What seems to be happening in the data is a moderation effect. Need for help moderates the relationship between wellbeing and the predictors. However, I think that the way the authors chose to present their data is clearer for readers, so I would not request a moderation analysis.
Conclusion: The article is interesting. It contributes to the literature and should be publishable after some rather minor additions, explanations, and clarifications.
Author Response
We are enclosing the revised manuscript of our paper "Friendly Residential Environments And Subjective Well-Being In Older People With And Without Help Needs” (ijerph-1980314). We are very grateful for the comments of the reviewers and believe that the manuscript has benefited greatly from their suggestions and feedback. Below, we respond in detail on how we have dealt with each of the comments and suggestions. The reviewer’s comments are in italic followed by our answers. In the paper the changes can be found as they are inserted with change tracker.
- The purpose of the article was well developed in the introduction.
- The sample was especially strong, being representative.
- However, no mention is make of what ethical procedures were followed in the research.
We have included a statement about the ethics protocol submission for this study.
- As well, the research was conducted during the beginning of COVID, and I recall that Spain was badly affected by this. This should be mentioned.
We have included a sentence and reference about the COVID-19 situation in Spain in the period the research was conducted.
- I would have appreciated a better development of the "need for help" variable in the introduction. It is mentioned but not defined, especially not supported as a dichotomy.
We have included a further development of the concept “need for help” in the introduction, as suggested.
- In Tables 1 and 2, the variables mentioned do not exactly match those listed in the Method. It would help if Table 1 was sub-classified according to categories in the Method. As well, variables were broken down and used individually rather than as a whole scale total.
The Tables 1 and 2 were reworked in order to include sub-classifications in order to match them with the variables listed in Methods section.
- In Table 2, the source of the R squared values needs to be clarified. The asterisks have no reference point. I take it the values being referred to are for the full models including the extra variables.
The sources of R squared values were clarified by indicating the asterisks close to the variables included in the final model.
- The authors discuss differences between regression models without performing any significance testing. In most cases where a confidence interval is provided, there is no difference among slope values. There is no evidence of added significant prediction provided for the demographic variables.
In comparing the two models, the authors can safely limit their discussion to variables the "were" and "were not" significant in either model.
I suspect the overall prediction was significantly stronger in the model for those with needs. An F value can be calculated to test this using the "predicted variance" in each model and the associated df.
We have included the significance testing values for each model, as suggested.
- It is perhaps rather obvious that those who need help will respond systematically to access and availability variabilities in judging their wellbeing. Before I broke my leg I never considered elevators in a two-storey building important to my wellbeing. The authors get to this point eventually.
- What seems to be happening in the data is a moderation effect. Need for help moderates the relationship between wellbeing and the predictors. However, I think that the way the authors chose to present their data is clearer for readers, so I would not request a moderation analysis.
Conclusion: The article is interesting. It contributes to the literature and should be publishable after some rather minor additions, explanations, and clarifications.
Round 2
Reviewer 1 Report
My primary concern is that the paper now states that 'no ethics review was submitted' to anyone. In our terms, this would not be acceptable for publication. For that reason, I must decline reviewing this paper or reject it for publication.
Author Response
We are grateful for the comment provided by the reviewer and we are aware we have not contextualised our response. In this case, we need to provide information about how ethics committees are regulated in Spain. In our country the ethics committee is governed under the Organic Law on Data Protection LOPD 3/2018 regulation. According to this regulation, anonymised datasets do not fall under the scope of application of the General Data Protection Regulation (GDPR), this being the case of the data presented in the work that was submitted for evaluation. However, if we can include or provide any information that may help to contextualise this in the manuscript, we would be very glad to prepare it in a proper way.
